# Geographical risk pattern and temporal trends in incidence of HPV-related cancers in northern Thailand: A population-based study

Patumrat Sripan[1], Donsuk Pongnikorn[2], Imjai Chitapanarux[3,4,5], Arunrat Tangmunkongvorakul[1], Karnchana Daoprasert[2], Linda Aurpibul[1], Narate Waisri[5], Puttachart Maneesai[5], Galyarath Wannavongs[6], Voravit Suwanvanichkij[1], Kriengkrai Srithanaviboonchai[1,7] *

1 Research Institute for Health Sciences, Chiang Mai University, Chiang Mai, Thailand, 2 Cancer Registry Unit, Lampang Cancer Hospital, Lampang, Thailand, 3 Northern Thai Research Group of Radiation Oncology (NTRG-RO), Faculty of Medicine, Chiang Mai University, Chiang Mai, Thailand, 4 Division of Radiation Oncology, Department of Radiology, Faculty of Medicine, Chiang Mai University, Chiang Mai, Thailand, 5 Chiang Mai Cancer Registry, Faculty of Medicine, Chiang Mai University, Chiang Mai, Thailand, 6 Nursing Department, Lampang Cancer Hospital, Lampang, Thailand, 7 Department of Community Medicine, Faculty of Medicine, Chiang Mai University, Chiang Mai, Thailand

* kriengkrai.s@cmu.ac.th

**Data Availability Statement:** All relevant data are within the article and its Supporting Information files.

## Abstract

### Background

The burden of HPV-related cancers in different regions worldwide varies according to several factors. This study aims to measure inequality in the risk of incidence of HPV-related cancers in term of geographical risk patterns in northern Thailand using a population-based cancer registry data.

### Methods

Trends in age-standardized HPV-related cancer incidence were calculated for the 2008–2017 time period. The Besag-York-Mollié model was used to explore the spatial distribution of the relative risk (RR) of HPV-related cancers at the district level. A higher RR reflects a larger disparity. The geographical risk pattern of the diseases in two periods, 2008–2012 and 2013–2017 were described and compared.

### Results

From 2008 to 2017, the incidence of oropharyngeal and anal cancers showed a slightly increased trend in males but remained stable in females, the incidence of vulvar, vaginal and penile cancers were stable while the incidence of cervical cancer decreased. The RR range was closer to 1 in the second period compared to the first period. This suggests a decrease in the disparities of incidence of cervical cancer. However, in some areas near the Thai-Myanmar border, the RR values remained high.

**Funding:** This research work was supported by Chiang Mai University. There was no grant numbers. The funders had no role in study design, data collection and analysis, decision to publish, or preparation of the manuscript.

**Competing interests:** The authors have declared that no competing interests exist.

## Conclusion

The incidence rate of most HPV-related cancers remained low and stable over the study period in northern Thailand. For the most common HPV-related malignancy, cervical cancer, the incidence rate steadily decreased but with marked geographic disparities, possibly reflecting health inequity especially in the border areas.

## Background

An estimated 15% of human cancers are caused by viral infections [1]. Human papillomavirus (HPV) is the most common oncogenic virus in low and medium Human Development Index (HDI) countries [2]. HPV accounts for approximately 600,000 cases of cancer of the cervix, oropharynx, anus, vulvar and penis worldwide. Overall, HPV is associated with more than 90% of anal and cervical cancers, about 70% of vaginal and vulvar cancers, 70% of oropharyngeal cancers and more than 60% of penile cancers [3]. HPV types 16 and 18 are the most common oncogenic viruses among the HPV subtypes known to cause cancer [4]. Cervical cancer is still one of the most common cancers in women in Southeast Asia [5].

The risk of HPV infection can be reduced through primary prevention, including safer sexual practices [6] and HPV vaccination, which is safe and effective [7]. Two prophylactic vaccines against HPV infection Cervarix® and Gardasil®, are available in Thailand. Both vaccines prevent HPV 16/18, and Gardasil® also prevents HPV types 6 and 11. Although, the HPV vaccine has been licensed in Thailand since 2007, it is only available primarily through private sources. With Thai public immunization programs have only been available for specific populations of school-age girls starting in 2018 and are not yet available for boys because of limited resources [8]. For people with HPV infection, secondary preventive measures can be implemented by early detection. Cervical cancer screening has been shown to reduce the incidence and mortality from the disease [9–11]. In Thailand, cervical cancer screening every five years is available for all Thai women aged 30–60 years under the Universal Health Care coverage (UHC) package.

The UHC has decreased the incidence of cervical cancer [12–14] and increased survival [15] since the establishment of national screening programs. The screening strategy using Papsmear and visual inspection with acetic acid (VIA) was started in 2005. After primary HPV 16/18 testing from cervical swabs every five years was found to be more cost-effective than cytology testing for cervical cancer in Thai women [16], the HPV testing for cervical screening was begun in 2020 as a pilot in some selected areas across the country. This method will replace conventional screening nationwide in 2022. However, there are currently no official screening guidelines for other HPV-related cancers in Thailand.

Recently, the International Agency for Research on Cancer (IARC) and the Catalan Institute of Oncology (ICO) reported the global magnitude of trends in HPV-related cancers, including cervical, anal, vulvar, vaginal, penile and oropharyngeal cancers. Cervical cancer was the most common HPV-related cancer and the other cancer types were rare [17]. Although, cervical cancer incidence has been decreasing worldwide, it remains a major health problem for women [18]. Some studies showed disparities in incidence and mortality rates of genital HPV-related cancers, with rural populations having higher rates than urban populations [19, 20]. Our study aims to measure inequality in the risk of incidence of HPV-related cancers including anal, cervical, oropharyngeal, penile, vaginal, and vulvar cancers in term of spatial risk patterns in different geographic areas of northern Thailand using population-based data.

## Material and methods

### Data

In this study, we used data from population-based cancer registries in the upper northern provinces of Thailand, which includes Mae Hong Son, Chiang Mai, Chiang Rai, Lampang, Lamphun, Phayao, Phrae and Nan. From these registries, data of all adults (age >15 years) from January 2008 to December 2017 with the International Classification of Diseases version 10 (ICD-10) diagnoses of cervical cancer (C53) which is the most common HPV-related cancer, anal cancer (C21) and oropharyngeal cancer (C10) which are the HPV-related cancers that occur in both males and females, and gender-specific cancer included vulvar cancer (C51) vaginal cancer (C52) and penile cancer (C60) was obtained from the registries. The population database from each province was used as the denominator for calculating age-standardized incidence rates (ASR), calculated for 5-years age intervals (starting from 15–19 to 80–84 and 85+) within each study area. The home address of each patient was recorded at the district level with 103 districts (Fig 1) using their postal code not linked to specific individuals. These aggregate data can be used as a representation of individual data under the assumption of homogeneity of demographic characteristics among the persons residing in the same geographical area.

The trends in ASR per 100,000 person-years of HPV-related cancers in the 2008–2017 time period were analyzed using the data from six cancer registries in the time period of 2008–2012 and eight cancer registries in the 2013–2017 time period because the Mae Hong Son and Nan cancer registries did not exist before 2013. The geographical risk pattern of HPV-related

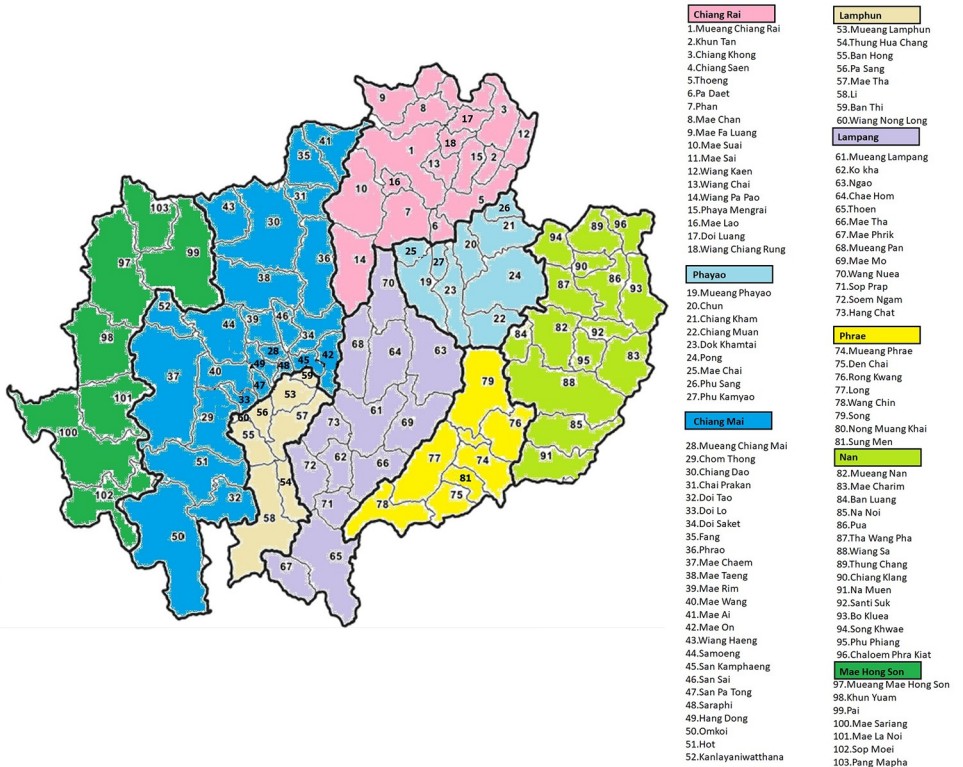

**Fig 1. The study area (upper northern Thailand).** The map in this figure was produced using Quantum Geographic Information System program (QGIS) version 3.4.15–1. Source of shapefile: https://data.humdata.org/dataset/geoboundaries-admin-boundaries-for-thailand.

cancers in 8 provinces, all upper northern Thailand, was described using the later 5-year periods (2013–2017). The temporal change by geographical areas between the two 5-years period 2008–2012 and 2013–2017, were described using data from six cancer registries, excluding Mae Hong Son and Nan. In the registries, more than 80% of the cancer diagnoses were morphologically verified (MV%) and less than 5% of the cancer diagnoses were from death certificate only (DCO%) in which the confirmed information could not be traced back. This indicates satisfactory data quality for the cancer registry data [21].

## Statistical analyses

In this study, the Besag-York-Mollié (BYM) model [22] was used to explore the spatial distribution for HPV-related cancer risk in northern Thailand. We estimated the relative risk (RR) of HPV-related cancer incidence for each district in upper northern Thailand and compared it to the incidence in the entire northern region of the country, which used as the baseline reference, and calculated 95% credible intervals (CrI). The RR were significantly higher than 1 when the 95% CrI were over 1. A map of the incidence patterns was then generated using the Quantum Geographic Information System program (QGIS) version 3.4.15–1 [23]. The shape-file used was from Global Administrative Database (geoBoundaries): an online, open license resource of the geographic boundaries of political administrative divisions [24].

This study was approved by the Human Experimentation Committee of Research Institute for Health Sciences, Chiang Mai University and the Research Ethics Committee of the Lampang Cancer Hospital.

## Results

### Study population

Our study analyzed data of patients diagnosed with HPV-related cancers over two time periods: 4,448 patients in 2008–2012 and 3,883 patients in 2013-2017.The proportion of anal, cervical, oropharyngeal, penile, vaginal, and vulvar cancers was 104 (2.3%), 3,930 (88.4%), 52 (1.2%), 205 (4.5%), 48 (1.1%), and 109 (2.4%), respectively, in the first time period, and 141 (3.6%), 3,229 (83.2%), 86 (2.2%), 224 (5.8%), 50 (1.3%), and 153 (3.9%), respectively, in the second time period. Quality of data indicators for the cancer registry and cancer sites are shown in S1 Table. The MV% ranged from 75 to 90 for cancer patients diagnosed in 2008–2012 and from 80 to 91 for cancer patients diagnosed in 2013–2017.

### Temporal trends in HPV-related cancers

The incidence of cervical cancer in the upper northern region of Thailand noticeably decreased from an ASR of 24.8 per 100,000 person-years in 2008 to an ASR of 13.2 per 100,000 person-years in 2017 (Fig 2).

During the entire study period, from 2008 to 2017, the ASR per 100,000 person-years for anal cancer did not change and remained stable at around 0.3 in both sexes (range: 0.25–0.48, 0.25–0.61 in males and 0.19–0.42 in females). The ASR per 100,000 person-years for oropharyngeal cancer did not change and remained stable at around 0.2 (range: 0.09–0.26, 0.16–0.40 in males and 0.3–0.14 in females). For more gender-specific HPV-related cancers, the ASR per 100,000 person-years remained stable at around 1.3 for penile (range: 0.99–1.42) cancer, 0.2 for vaginal cancer (range: 0.15–0.31), and 0.6 for vulvar cancer (range: 0.50–0.77) (Fig 3).

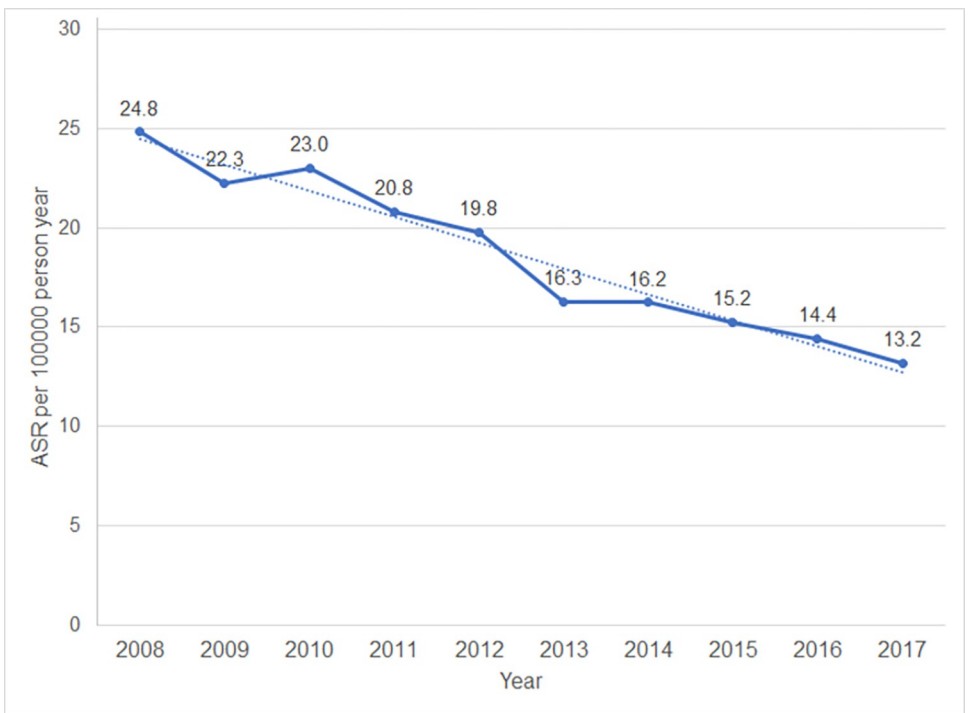

**Fig 2. Trends in the incidence of cervical cancer in upper northern Thailand.**

## Geographical risk pattern of HPV-related cancers

**Cervical cancer.** The geographic variations in RR of cervical cancer in the time period of 2013–2017 in upper Northern Thailand by district is shown in Fig 4. The areas with RR >1 were shown in S2 Table (range from 1.05 to 2.23). The incidence rates of cervical cancer were

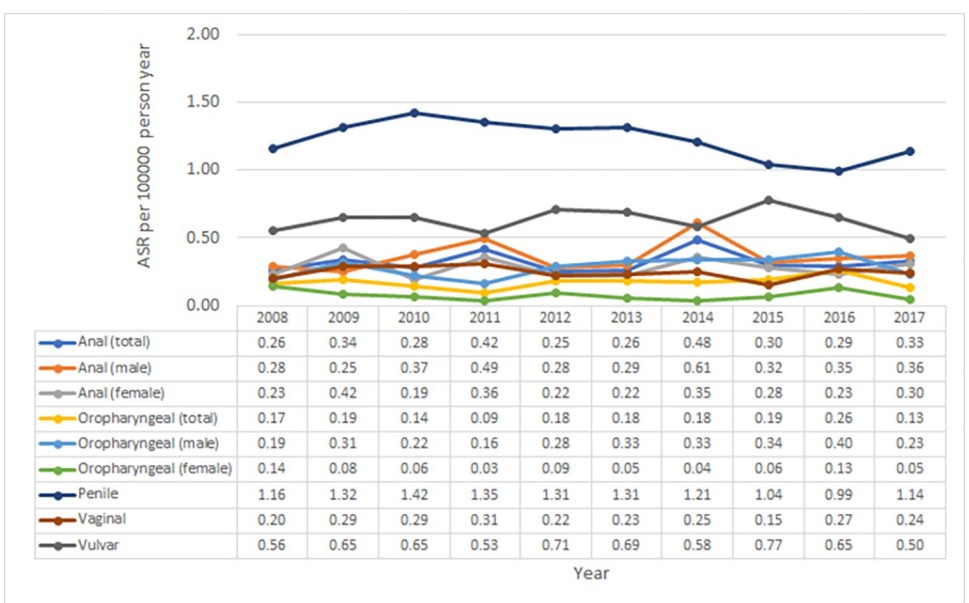

| | 2008 | 2009 | 2010 | 2011 | 2012 | 2013 | 2014 | 2015 | 2016 | 2017 |
|---|---|---|---|---|---|---|---|---|---|---|
| Anal (total) | 0.26 | 0.34 | 0.28 | 0.42 | 0.25 | 0.26 | 0.48 | 0.30 | 0.29 | 0.33 |
| Anal (male) | 0.28 | 0.25 | 0.37 | 0.49 | 0.28 | 0.29 | 0.61 | 0.32 | 0.35 | 0.36 |
| Anal (female) | 0.23 | 0.42 | 0.19 | 0.36 | 0.22 | 0.22 | 0.35 | 0.28 | 0.23 | 0.30 |
| Oropharyngeal (total) | 0.17 | 0.19 | 0.14 | 0.09 | 0.18 | 0.18 | 0.18 | 0.19 | 0.26 | 0.13 |
| Oropharyngeal (male) | 0.19 | 0.31 | 0.22 | 0.16 | 0.28 | 0.33 | 0.33 | 0.34 | 0.40 | 0.23 |
| Oropharyngeal (female) | 0.14 | 0.08 | 0.06 | 0.03 | 0.09 | 0.05 | 0.04 | 0.06 | 0.13 | 0.05 |
| Penile | 1.16 | 1.32 | 1.42 | 1.35 | 1.31 | 1.31 | 1.21 | 1.04 | 0.99 | 1.14 |
| Vaginal | 0.20 | 0.29 | 0.29 | 0.31 | 0.22 | 0.23 | 0.25 | 0.15 | 0.27 | 0.24 |
| Vulvar | 0.56 | 0.65 | 0.65 | 0.53 | 0.71 | 0.69 | 0.58 | 0.77 | 0.65 | 0.50 |

**Fig 3. Trends in the incidence of non-cervical HPV-related cancers.**

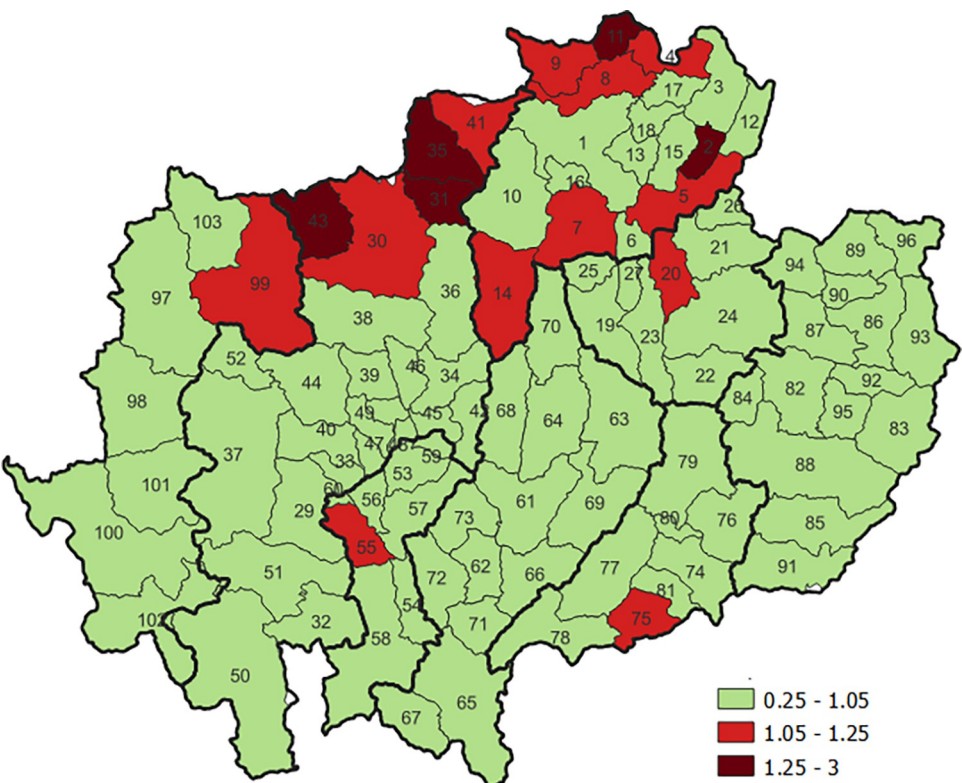

**Fig 4. Geographic variation in relative risk of cervical cancer by district in upper northern Thailand.** The map in this figure was produced using QGIS version 3.4.15–1. Source of shapefile: https://data.humdata.org/dataset/geoboundaries-admin-boundaries-for-thailand.

significantly higher than the average of the incidence in Northern Thailand (baseline reference), in order from high (dark red) to low (green) value of RR were found in districts of Chiang Mai Province, Chiang Rai Province, Phrae Province, Mae Hong Son Province, Phayao Province and Lamphun Province (Fig 4).

The incidence of cervical cancer compared to the baseline reference was significantly higher in 21 districts in 2008–2012 (Fig 5A) and remained higher in 16 districts in 2013–2017 (Fig 5B) (S3 Table). These were districts bordering Myanmar (Fig 5).

**Non-cervical HPV-related cancers.** The geographical risk pattern of non-cervical HPV-related cancers in the 2013–2017 time period are shown in Fig 6 and Fig 7. The incidence rate of HPV related cancers was significantly higher than the baseline reference. The RR were significantly higher than 1 for anal cancer in a specific district of Lamphun Province (RR = 1.26, 95% CrI:1.03–1.51) and a district of Chiang Mai Province (RR = 1.24, 95% CrI:1.02–1.49) (Fig 6A), and significantly higher than 1 for oropharyngeal cancer in a specific district of Chiang Mai Province (RR = 1.34, 95% CrI:1.02–1.71) (Fig 6B).

The incidence rate of gender-specific HPV related cancers varied by geographical area. For HPV related cancer in men, the incidence rate of penile cancer was significantly higher than the baseline reference in the districts of Chiang Rai Province, Mae Hong Son Province, Chiang Mai Province, Lampang Province and Chiang Rai Province (Fig 7A) with RR ranging from 1.22 to 8.63 (S2 Table).

The incidence rate of HPV related cancer in women was significantly higher than the baseline reference for vaginal cancer in some districts of Phayao province and some districts in

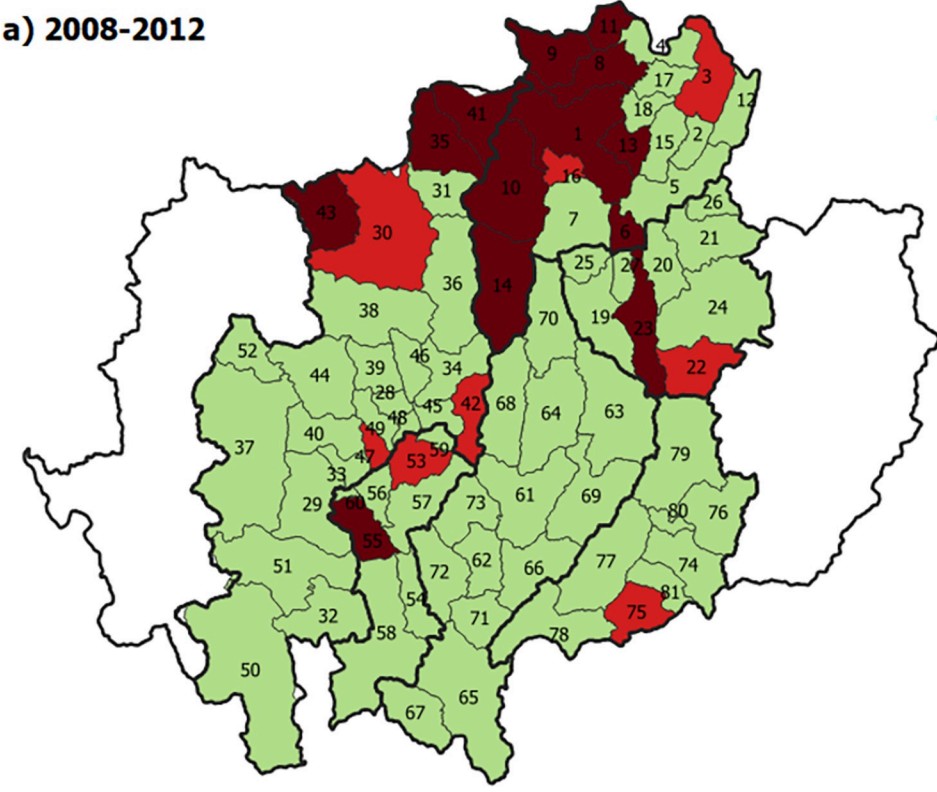

a) 2008-2012

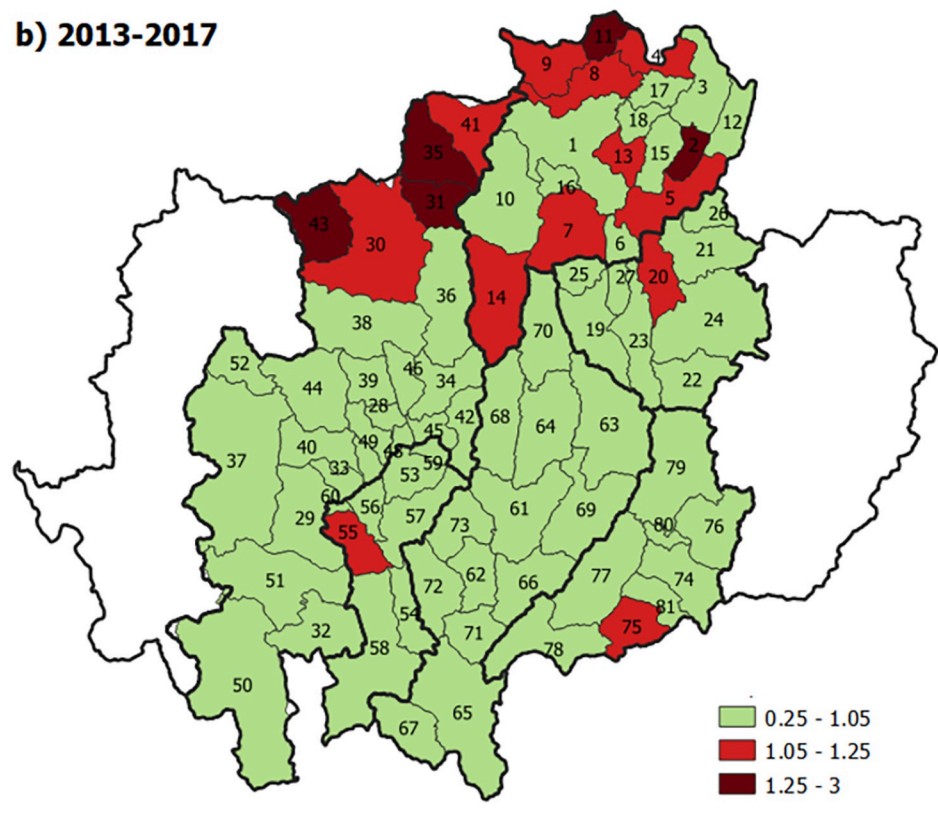

b) 2013-2017

0.25 - 1.05
1.05 - 1.25
1.25 - 3

**Fig 5. Temporal changes in relative risk of cervical cancer by geographic area in northern Thailand.** The map in this figure was produced using QGIS version 3.4.15–1. Source of shapefile: https://data.humdata.org/dataset/geoboundaries-admin-boundaries-for-thailand.

Chiang Rai province (Fig 7B) with RR significantly higher than 1 (1.56 to 2.70, S2 Table), and significantly higher than the baseline reference for vulvar cancer in districts of Chiang Mai Province and Chiang Rai Province (RR 1.29–1.73, S2 Table) (Fig 7C).

## Discussion

The trends in incidence of HPV-related cancers in northern Thailand varied by cancer type. From 2008 to 2017, the incidence of oropharyngeal and anal cancers, slightly increased from 2008 to 2017 in men but remained stable in women, the incidence of the gender-specific HPV-related cancers (cervical, vulvar and vaginal cancers in females and penile cancer in males) excluding cervical cancer remained stable and the incidence of cervical cancer, the most common HPV-related cancer, decreased (Fig 1). Several studies in Thailand have reported incidence trends of cervical cancer, but trends of other HPV-related cancers have been less frequently evaluated [14, 15, 25–27]. Our study is one of only a few studies reporting the magnitude of HPV-related cancer incidence in Thailand. The incidence trends in HPV-related cancers in our study conformed to results reported by IARC and ICO which used data from 2008 to 2012 [17]. Similarly, in the USA, overall incidence rates increased for oropharyngeal, anal and vulvar cancers, decreased for cervical and vaginal cancers, and remained stable for penile cancer. The decline in cervical cancer in the US population from 1999 to 2015 represents a trend since the 1950s which is largely attributable to cancer screening [28]. Decreases in cervical cancer incidence have also been found in other low- and middle-income countries [18]. Similar trends have been observed in other studies in the Thai population, in which the incidence of cervical cancer decreased [12–14] while survival increased [15].

The decreased incidence of cervical cancer likely reflects the overall impact of national policies on HPV-related cancer. Our previous study in Chiang Mai province showed a reduction of incidence and mortality rates of cervical cancer mainly in urban areas but no significant decrease in more rural areas [20]. Thus, spatial analyses of HPV-related cancers may help us to better understand the geographic variations of the disease burden and provide insight into important public health determinants of the incidence of HPV-related cancers. This data will be essential for prevention and control of these malignancies, particularly cervical cancer, which is the most common HPV-related cancer.

This study demonstrates significant geographical disparities in the incidence of all HPV-related cancers in the later periods (2013–2017). Since most of the HPV-related cancers were rare except for cervical cancer, we could only describe temporal changes in the relative risk of cervical cancer by geographic area. The inequality in the decrease of cervical cancer incidence in the later period is reflected in the RR being closer to 1. Our results show a decrease in the number of areas where the incidence rates were significantly higher than the average incidence in this region of Thailand in the later period (2013–2017) compared to the previous period (2008–2012). That is may be a result of the combination of the prevention program and national screening program in Thailand under the UHC. However, the incidence is still high in some areas. This trend happened the same way in southern Thailand and Costa Rica, the other part of the world where the incidence rates of cervical cancer decreased over time but with the rates at some border areas remaining high compared with the rest of the study area [27, 29]. These studies suggest that populations in border regions of the country should be prioritized for cervical cancer control.

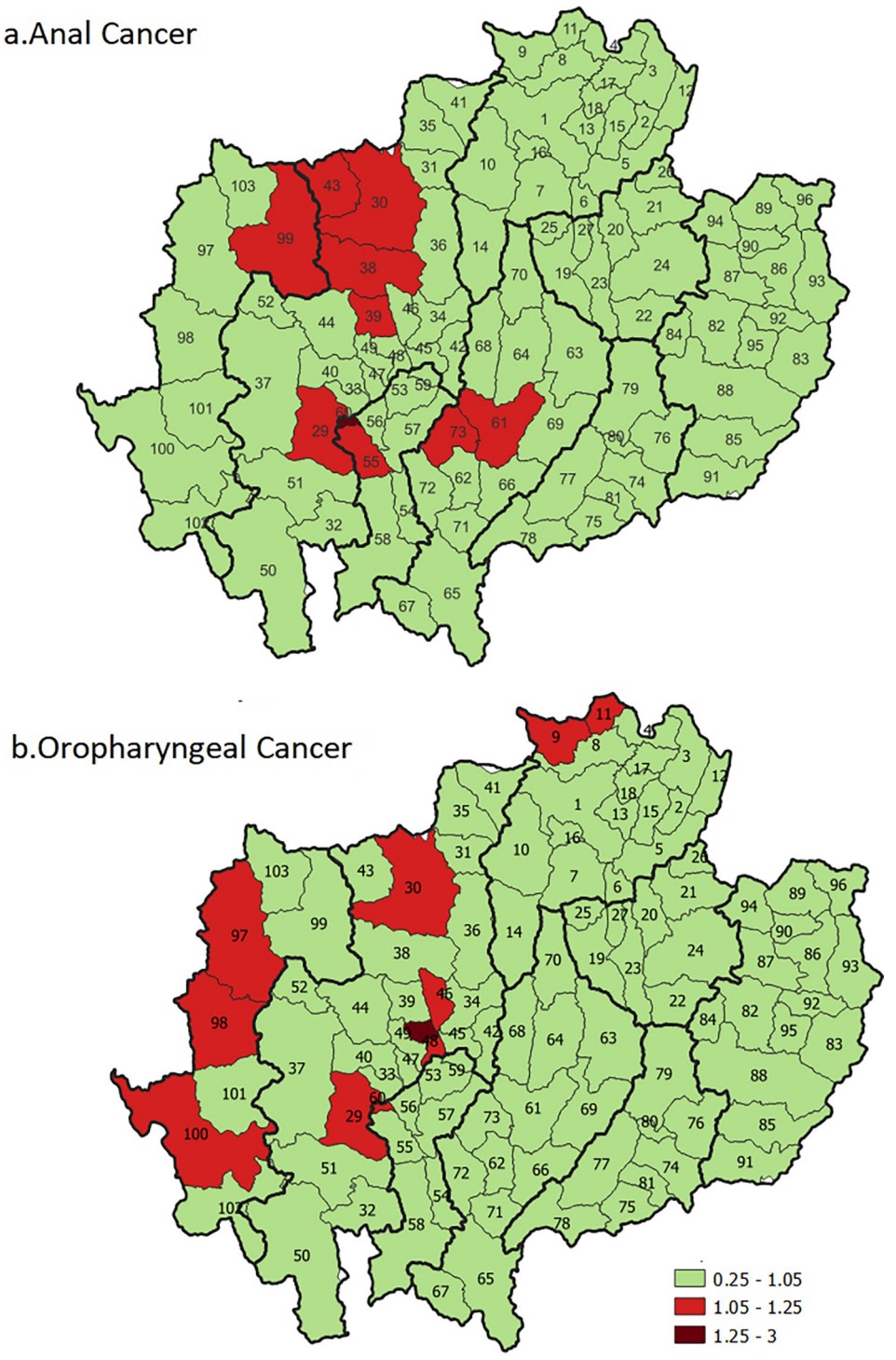

**Fig 6.** Geographic variations in the relative risk of a) anal cancer and b) oropharyngeal cancer. The map in this figure was produced using QGIS version 3.4.15–1. Source of shapefile: https://data.humdata.org/dataset/geoboundaries-admin-boundaries-for-thailand.

Our spatial analyses of HPV-related cancer incidence in the more recent period (2013–2017) showed that the disease was unevenly distributed and varied significantly by

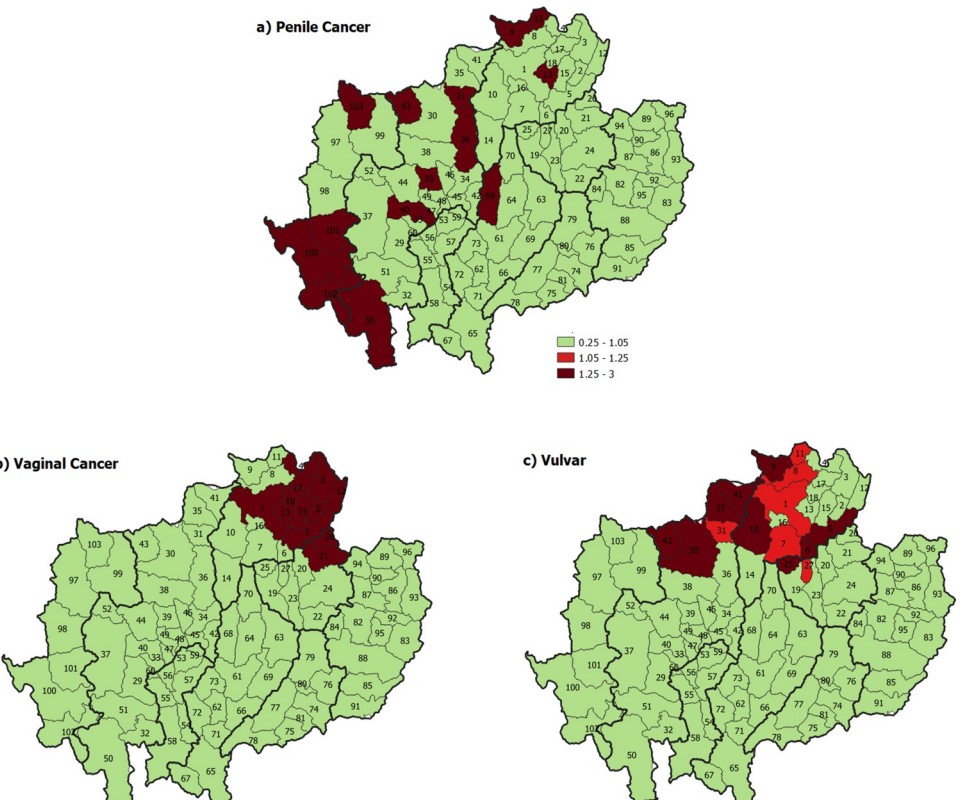

**Fig 7. Geographical risk pattern of gender-specific HPV-related cancers.** a) penile cancer, b) vaginal cancer, and c) vulvar cancer. The map in this figure was produced using QGIS version 3.4.15–1. Source of shapefile: https://data.humdata.org/dataset/geoboundaries-admin-boundaries-for-thailand.

geographical area. A higher risk of cervical, vaginal, vulvar, and penile cancers was found in many areas near the border with Myanmar. This may be related to documented challenges in transnational mobility and associated health care access as reported in a qualitative study by medical and non-medical hospital staff responsible for implementing facility-level policies and providing health services for transnational populations [30]. This could explain the disparity in access to HPV prevention strategies, including screening and vaccination under the UHC, in the populations along the border areas because these prevention strategies only cover some categories of Thai citizens and not migrant people. The highest cervical cancer RR are likely to be in border areas, primarily along the Chiang Rai and Chiang Mai border but not the Mae Hong Son border. This is might be due to the economically-active border and the large scale legal transborder migration in Chiang Rai and Chiang Mai. The small number of checkpoints in upper northern Thailand are located in districts of Chiang Rai province, Mae Sai District and some in Chiang Saen District. In some areas, there were also problems from STD in the past because of covert commercial sex [30].

Prevention programs include avoiding unprotected sex and educating people about how safe sexual activities can reduce STDs. During the last decade, the Thai government has made efforts on these programs in order to decrease STD incidence. This might also result in partially reducing HPV infection and HPV-related cancers in Thailand. HPV infection and the risk of subsequent related malignancies can also be reduced through HPV immunization programs targeting younger populations, especially before sexual debut [7]. The effect of HPV

immunization on prevalence of HPV infection might be seen in the next generation. The Thai national immunization program has included HPV vaccination for all Thai national female students in grade 5 since 2018 under the UHC with full reimbursement [8]. To reduce disparities in the incidence of HPV-related cancers, this primary prevention strategy may be an effective and practical strategy in the near future and the government should make more efforts to promote this strategy for populations in particular in the border areas.

One limitation of our study is that the population-based data set did not include potentially confounding variables such as HPV infection status, and tobacco and alcohol intake [31–35]. The attributable risk for HPV related cancer largely varies according to cancer type, strong association with cervical cancers and less association with oropharyngeal cancer. The high incidence of some HPV-related cancers, particularly oropharyngeal cancer, in Chiang Mai province, Chiang Rai province and Mae Hong Son province was possibly related to high prevalence of smoking in those areas in the decades before Thailand inducted a national tobacco control measure in 1989 [36]. Potential confounding variables including lifestyle and cultural differences which have direct impacts on the incidence of HPV-related cancers should also be addressed.

HPV prevalence is the most important factor that should be measured. This information has not been available for northern Thailand. The inequality in knowledge about HPV prevalence was also found. A number of studies reported various proportions of HPV infection in patients diagnosed with different type of HPV related cancers. Nevertheless, mostly the study population were living in city areas [37–43]. Only a few studies reported prevalence of HPV in the general Thai population. In women attending a cervical cancer screening mobile unit in Lampang, the prevalence of HPV DNA was 5.4% [44]. Another study found that the prevalence of HPV DNA was higher among women from the northern region than the south (9.1% vs 3.9%) [45]. Primary HPV testing has been included under the UHC since 2020 and will cover the entire country in 2022. This data will allow us to better understand the geographic difference of HPV prevalence in Thailand. Future research should be conducted to explain the reasons behind the inequality in incidence of HPV-related cancer in the era of HPV testing under the UHC.

Another limitation of our study is that the geographical pattern of RR of HPV-related cancer could be only evaluated in the second period of our study because the two newest population-based cancer registries, for Mae Hong Son and Nan, only existed in the second period. Therefore, we could describe the temporal changes in relative risk of cervical cancer by geographic area of Northern Thailand using the data from only six cancer registries (Fig 3). S1 Table shows that by combining data from all cancer registries in Northern Thailand, the data quality was satisfactory. Satisfactory is considered to mean MV% of more than 75% for all cancer sites and DCO% not exceeding 5% [21].

## Conclusion

In conclusion, the trends in incidence were slightly increasing for oropharyngeal and anal cancers in males, stable and low for vulvar, vaginal cancers and penile cancer, and decreasing for cervical cancer in our study in the northern Thai population. Although, the cervical cancer incidence rates steadily decreased in the later period (2013–2017) compared to the earlier period (2008–2012), the incidence in some specific areas remained high, particularly in rural communities along the Thai-Myanmar border in Chiang Mai and Chiang Rai provinces. This suggests that significant gaps exist in health care coverage of these areas, particularly access to routine cervical cancer screening for women. Future research should address the vulnerabilities contributing to significant health disparities in these populations. In the near future, HPV

vaccination should be an important strategy to reduce these disparities and incidence of HPV related-cancer. We encourage health authorities to scale up the coverage of HPV vaccination in the young population before they become sexually active particularly in high-risk areas.

## Supporting information

**S1 Table. Data quality indicators.**
(DOCX)

**S2 Table. Areas with relative risk (RR) significantly higher than 1 by cancer type, 2013–2017.**
(DOCX)

**S3 Table. Areas with relative risk (RR) significantly higher than 1 for cervical cancer by time period.**
(DOCX)

**S1 Data.**
(XLSX)

## Acknowledgments

We would like to thank the Bureau of Registration Administration, Ministry of Interior, for providing death certificate data for cancer registry.

## Author Contributions

**Conceptualization:** Patumrat Sripan, Kriengkrai Srithanaviboonchai.

**Data curation:** Donsuk Pongnikorn, Imjai Chitapanarux, Karnchana Daoprasert, Narate Waisri, Puttachart Maneesai, Galyarath Wannavongs.

**Formal analysis:** Patumrat Sripan.

**Investigation:** Patumrat Sripan, Donsuk Pongnikorn, Imjai Chitapanarux, Arunrat Tangmunkongvorakul, Karnchana Daoprasert, Linda Aurpibul, Kriengkrai Srithanaviboonchai.

**Validation:** Donsuk Pongnikorn, Imjai Chitapanarux, Arunrat Tangmunkongvorakul, Linda Aurpibul, Narate Waisri, Puttachart Maneesai, Galyarath Wannavongs, Voravit Suwanvanichkij, Kriengkrai Srithanaviboonchai.

**Writing – original draft:** Patumrat Sripan, Voravit Suwanvanichkij, Kriengkrai Srithanaviboonchai.

**Writing – review & editing:** Patumrat Sripan, Donsuk Pongnikorn, Imjai Chitapanarux, Arunrat Tangmunkongvorakul, Karnchana Daoprasert, Linda Aurpibul, Narate Waisri, Puttachart Maneesai, Galyarath Wannavongs, Voravit Suwanvanichkij, Kriengkrai Srithanaviboonchai.

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
