## [Decision Letter · Decision Letter 0]

7 Apr 2022

PONE-D-22-0405Inequality in the Risk of Incidence of HPV-related Cancers in Northern Thailand Population-Based Spatial AnalysisPLOS ONE

Dear Dr. Srithanaviboonchai,

Thank you for submitting your manuscript to PLOS ONE. After careful consideration, we feel that it has merit but does not fully meet PLOS ONE’s publication criteria as it currently stands. Therefore, we invite you to submit a revised version of the manuscript that addresses the points raised during the review process.

Critically it is necessary to address methodological concerns and ambiguities raised by the reviewers.

The data presentation can also be improved as well as the overall language of the manuscript.

We look forward to receiving your revised manuscript.

Kind regards,

Ivan Sabol

Academic Editor

PLOS ONE

Journal Requirements:

(We would like to thank the Bureau of Registration Administration, Ministry of Interior, for providing death certificate data for cancer registry. This research work was supported by Chiang Mai University.)

(The funders had no role in study design, data collection and analysis, decision to publish, or preparation of the manuscript.)

5. We note that Figures 3, 4, 5, and 6 your submission contain map images which may be copyrighted. All PLOS content is published under the Creative Commons Attribution License (CC BY 4.0), which means that the manuscript, images, and Supporting Information files will be freely available online, and any third party is permitted to access, download, copy, distribute, and use these materials in any way, even commercially, with proper attribution. For these reasons, we cannot publish previously copyrighted maps or satellite images created using proprietary data, such as Google software (Google Maps, Street View, and Earth). For more information, see our copyright guidelines: http://journals.plos.org/plosone/s/licenses-and-copyright.

a. You may seek permission from the original copyright holder of  Figures 3, 4, 5, and 6  to publish the content specifically under the CC BY 4.0 license.  

Additional Editor Comments:

P5 L117 what was the rationale for selecting those 2 particular time periods? It might be worth highlighting that both are 5 year periods.

P5 L120 what is meant by “morphologic verification”? later in the manuscript it appears that this was abbreviated to MV%? If so, please clarify

P5 L121 The part about diagnosis from death certificate is also a bit unclear and should be better explained.

P5 L128 the methods mention that credible intervals (CrI) were calculated. However, this information is never shown in the manuscript. Consider including the calculated information at least in supplementary tables

P5 L129 Some reference, version number, or at least a link to the QGIS program should be provided. Are there some specific options used that would allow others to analyze the data in the same way?

P5 L138 the cancer types and numbers provided should be sorted in some meaningful order. Ie HPV involvement, descending numbers, alphabetical order or some other consistent order. The same order should be used in Supplementary Table 1 as well as throughout the manuscript and figures

P6 L148 the sentence “The RR ranges from 1.05 to 2.23“ might be factually inconsistent with figure 2 since it is obvious that some regions pictured have RR <1

P6 L149 and 153 (and elsewhere) Was some statistical test performed to compare incidences between study regions or time periods, if so some p value and test used should be shown? If not, the use of “significantly” might be misleading and should be replaced with “noticeably” or some other similar word which could not be misunderstood for statistical significance.

Can the underlying data of Figures 1/4 be supplied as supplemental table (and include the same information for other cancer types?

Figures 1 and 4 should be combined since they present very similar data, at least as subpanels

Can Figure 2/3/5/6 colour scales be changed so that RR less than one is in shades of green (good) and RR above 1 in shades of red (bad) or some other 2 color scale combination. Currently it is difficult to spot areas with RR~1 and maybe areas with lower RR are not emphasized as such and appear more like a „baseline“ instead of "less than average"

Instead of repeatedly listing the names of 103 provinces in each figure 2/3/5/6 please make a supplementary table with all the names and refer to this list from each figure legend. The same supplementary table could/should include some of the underlying data (ie. actual number of cases found for each cancer type as well as the actual calculated RR number, population used for calculation, etc..

Subheadings could be emphasized by bold font to be better distinguishable from main text.

First line paragraph indents are inconsistently used (ie. P7 L173 vs 179)

The language needs some revision and proofreading with some particular points highlighted below

P3 L62 „cancers cause by viral infections“ should be „are caused by“

P3L83 „and increase survival“ should be „and increased survival“

P4 L100 typo “oropharyngeal, virginal, vulvar,“ should be vaginal

P5 L138 typo space „48(1.1%)“

P9 L226 lack of commas „cervical vaginal vulvar and“

P9 L227 grammar „This may relevant to the limited“

P9 L233 grammar „some categories Thai citizens“

Reviewers' comments:

Reviewer's Responses to Questions

**Comments to the Author**

1. Is the manuscript technically sound, and do the data support the conclusions?

Reviewer #1: Partly

Reviewer #2: Yes

Reviewer #3: Yes

2. Has the statistical analysis been performed appropriately and rigorously? 

Reviewer #1: Yes

Reviewer #2: Yes

Reviewer #3: Yes

3. Have the authors made all data underlying the findings in their manuscript fully available?

Reviewer #1: No

Reviewer #2: No

Reviewer #3: Yes

4. Is the manuscript presented in an intelligible fashion and written in standard English?

Reviewer #1: Yes

Reviewer #2: Yes

Reviewer #3: Yes

5. Review Comments to the Author

Reviewer #1: This manuscript reports trends in age-standardized HPV-related cancer incidence over the

time period from 2008-2017. I have below comments and questions.

What’s “MV%”, ASR and DCO%? Please define acronyms the first time they appear in the text.

The screening strategy using Pap-smear and visual inspection with acetic acid (VIA) has started in 2005. Why separate the data into two time periods as 2008-2012 and 2013-2017?

In the results, please report calculated 95% credible intervals.

In Figures 5 and 6, which time period was presented for the incidences by province?

Page 169-172, From figure 5, it has no information for sex. Why do you say that the incidence rate of HPV related cancers in “both sexes” were significantly higher than the average incidence in Northern Thailand in a specific district of Lamphun Province?

Reviewer #2: I found this paper an interesting one, week designed conducted and analyzed. I have few concerns:

1) In my opinion the authors should further stress the concept that the attributable risk for HPV related cancer largely varies according to cancer type -it is 100% only for cervical cancer. Therefore any interpretation must take into account the other risk factors -e.g., smoking for oral cavity.

2) The English text needs revision

Reviewer #3: Thank you for an interesting submission that provides useful data for the readership. I have a few suggestions that would improve the manuscript:

1. Title - the use of spatial analysis in the title is a little confusion - I would suggest 'Geographical and Temporal trends in HPV-related cancers in Northern Thailand' as an alternative

2. Results - I would separate out temporal trends in incidence and then geographical. When presenting the temporal trends in the text I think you should be more explicit that there was no increase over time (then present the range). At first glance it appears that you say there is no increase and then present the ASR that appears to rise. The figures help of course.

3. Results - Figure of ASR - I would have a single figure with all cancer types (as opposed to cervical cancer separate)

4. Results - geographical spread figure(s) - these are going to be very difficult to read on the page - suggest pick out the key one or two and put the rest in supplementary material.

5. Discussion - para 3 - sentence ' This may relevant to the limited to the domain....' this is poorly written and would be better simplified e.g. 'This may be related to documented challenges in transnational mobility and associated health care access'.

6. PLOS authors have the option to publish the peer review history of their article (what does this mean?). If published, this will include your full peer review and any attached files.

Reviewer #1: No

Reviewer #2: No

Reviewer #3: No

---

## [Author Response · Author response to Decision Letter 0]

5 May 2022

Please find the attached file named "Response to Reviewers".

---

## [Decision Letter · Decision Letter 1]

15 Jun 2022

Geographical Risk Pattern and Temporal Trends in Incidence of HPV-related Cancers in Northern Thailand: A Population-Based Study

PONE-D-22-04051R1

Dear Dr. Srithanaviboonchai,

We’re pleased to inform you that your manuscript has been judged scientifically suitable for publication and will be formally accepted for publication once it meets all outstanding technical requirements.

Kind regards,

Ivan Sabol

Academic Editor

PLOS ONE

Additional Editor Comments (optional):

Reviewers' comments:

Reviewer's Responses to Questions

**Comments to the Author**

1. If the authors have adequately addressed your comments raised in a previous round of review and you feel that this manuscript is now acceptable for publication, you may indicate that here to bypass the “Comments to the Author” section, enter your conflict of interest statement in the “Confidential to Editor” section, and submit your "Accept" recommendation.

Reviewer #1: All comments have been addressed

Reviewer #2: All comments have been addressed

Reviewer #3: All comments have been addressed

2. Is the manuscript technically sound, and do the data support the conclusions?

Reviewer #1: (No Response)

Reviewer #2: Yes

Reviewer #3: Yes

3. Has the statistical analysis been performed appropriately and rigorously? 

Reviewer #1: (No Response)

Reviewer #2: Yes

Reviewer #3: Yes

4. Have the authors made all data underlying the findings in their manuscript fully available?

Reviewer #1: (No Response)

Reviewer #2: Yes

Reviewer #3: Yes

5. Is the manuscript presented in an intelligible fashion and written in standard English?

Reviewer #1: (No Response)

Reviewer #2: (No Response)

Reviewer #3: Yes

6. Review Comments to the Author

Reviewer #1: (No Response)

Reviewer #2: The authors had revised the manuscript in accordance with all suggestions. Now it can be published without further revisions.

Reviewer #3: (No Response)

7. PLOS authors have the option to publish the peer review history of their article (what does this mean?). If published, this will include your full peer review and any attached files.

Reviewer #1: No

Reviewer #2: **Yes: **Diego Serraino, MD

Reviewer #3: No

---

## [Editor Report · Acceptance letter]

20 Jun 2022

PONE-D-22-04051R1 

Geographical Risk Pattern and Temporal Trends in Incidence of HPV-related Cancers in Northern Thailand: A Population-Based Study 

Dear Dr. Srithanaviboonchai:

I'm pleased to inform you that your manuscript has been deemed suitable for publication in PLOS ONE. Congratulations! Your manuscript is now with our production department. 

Kind regards, 

on behalf of

Dr. Ivan Sabol 

Academic Editor

PLOS ONE